# Independent fitness consequences of group size variation in Verreaux's sifakas
Peter M. Kappeler ●[1,2] ✉ & Claudia Fichtel ●[1]

The costs and benefits of group living are also reflected in intraspecific variation in group size. Yet, little is known about general patterns of fitness consequences of this variation. We use demographic records collected over 25 years to determine how survival and reproductive success vary with group size in a Malagasy primate. We show that female reproductive rates of Verreaux's sifakas (*Propithecus verreauxi*) are not affected by total group size, but that they are supressed by the number of co-resident females, whereas mortality rates are significantly higher in larger groups. Neither annual rainfall nor the adult sex ratio have significant effects on birth and death rates. Hence, these sifakas enjoy the greatest net fitness benefits at small, and not the predicted intermediate group sizes. Thus, independent fitness proxies can vary independently as a function of group size as well as other factors, leading to deviations from optimal intermediate group sizes.

Life in permanent groups has evolved repeatedly throughout the history of the animal kingdom, and the size, composition as wel as social structure of animal societies today varies widely among and within species[1]. Similar selection pressures are thought to govern both interspecific evolutionary transitions in social organisation and intraspecific variation in group size, which varies as a function of variation in rates of reproduction, survival, immigration and emigration in different age and sex categories[2–6]. Only a comparative study across 8 species of dolphins and porpoises found no general and consistent relation between group size and other variables within and across species[7]. However, whereas the general factors favouring the evolution of group living have been studied extensively during the early years of behavioural ecology[8–11], the drivers as well as the actual direct and indirect fitness consequences of intraspecific variation in social phenotypes remain less studied, even though they can be variable across species, populations and groups within species, sexes, age and dominance categories, years and environmental conditions[12–15], among other things because members of different categories vary in their abilities to control group size.

This variation in the links between sociality and fitness is distributed unevenly between two core components of social systems, i.e., social structure and social organisation[16], First, as in humans[17–19], studies of several species of Old World monkeys, but also of ungulates, cetaceans, hyraxes, rabbits and meerkats, revealed that various aspects of social structure, such as social status and integration, are positively associated with survival, benefitting individuals with strong social bonds or better social network connections[20–24]. Only studies of yellow-bellied marmots did not consistently find associations between social structure and fitness measures[25,26].

Second, variation in social organisation, and group size in particular, is associated with a wide array of fitness measures, though the effects are heterogeneous within and across species and rarely correspond with theoretical predictions about an optimal group size. For example, in one study of African wild dogs, groups size had no effect on survival[27], whereas adult survival decreased with pack size in another study[28]. In meerkats, mortality was higher in smaller groups[29], and in spotted hyenas, the size of one clan was positively associated with reproduction, but not with subsequent recruitment[30]. In rodents, survival and net reproductive rates increased with increasing group size, but decreased in the largest groups in yellow-bellied marmots[31], whereas in degus, adult female and offspring survival were not influenced by group size[32]. In prairie voles, juveniles survived better in groups of intermediate sizes, but adult survival was independent of group size[33]. Across primate studies, there appears to be a more uniform pattern characterised by fitness costs for individuals living in larger groups, as indicated by longer inter-birth intervals, lower fertility, reduced juvenile survival, or delayed sexual maturity[34–38].

Another set of studies analysed group size variation while simultaneously considering additional variables, because fluctuations in resources and other factors may contribute to the maintenance of the distributions of groups sizes around a presumed intermediate optimum[39,40]. In superb starlings, for example, group size was positively correlated with adult survival, but only for males in wet years[41], implicating an interaction between environmental conditions and sex modulating the group size-fitness link. Similarly, individual reproductive success of anis was higher in large groups than in small groups in wet years, whereas the opposite pattern was found in dry years[39], suggesting that fluctuating selection may maintain variation in

[1]Behavioral Ecology & Sociobiology Unit, German Primate Center, Leibniz Institute for Primate Research, Kellnerweg 4, 37077 Göttingen, Germany. [2]Department of Sociobiology/Anthropology, Johann-Friedrich-Blumenbach Institute of Zoology and Anthropology, University Göttingen, Kellnerweg 6, 37077 Göttingen, Germany. ✉e-mail: pkappel@gwdg.de

group size. In giraffes, in contrast, group size was much more important in predicting survival than social network metrics or any environmental factor[42]. Finally, a study of prairie voles[33], as well a recent experimental study in ostriches[43] demonstrated that the divergent reproductive interests of the sexes, as reflected by different adult sex ratios (ASR), can also promote different optimal group sizes.

Thus, in order to optimise fitness costs and benefits associated with different current group sizes, there should generally be stabilising selection for intermediate-sized groups[44–46], but the costs and benefits of small and large group sizes may vary locally with environmental and social factors, so that group size variation persists and the optimal group size does not necessarily coincide with the central tendency of group sizes. In particular, selection for different group sizes may well vary between members of different age, sex and dominance categories – as will their capacity to control it. The general pattern of fitness consequences remains relatively poorly understood, however, because different co-variates as well as short- and long-term fitness measures have been used in different studies. Therefore, explaining the extent, drivers and consequences of group size variation remains a key problem in social evolution, and data from additional taxa that explicitly consider additional social and environmental factors in shaping fitness are required for further insights.

The general aim of this study was therefore to examine fitness consequences of group size variation in a representative of an independent primate radiation from Madagascar–Verreaux's sifakas (*Propithecus verreauxi*). To put group size effects into the broader context of other recent studies, we also considered variation in ASR as well as interannual variation in rainfall and food availability as potential drivers of fitness variation. ASR variation, which is maximal in small groups[47], has recently been shown to have pronounced effects on reproductive strategies[48,49] as well as group size[43]. Rainfall, which is correlated with food availability, and, hence, ultimately condition, survival and reproduction, has been shown in a study of sympatric mouse lemurs to have varied over the last 30 years[50]. To understand variation in group size it is also important to understand the relative roles of reproduction, survival, immigration and emigration—and the extent to which individuals control them. Because both reproductive effort and mortality are known to be age-dependent[51], we therefore also controlled for the effects of age. Furthermore, because a previous study suggested that female competition might impact reproductive rates in sympatric redfronted lemurs[52], which also live in small groups with philopatric females, we also examined effects of variation in the number of adult females. Moreover, this study also contributes much needed comparative data on absolutely small group sizes, where per capita cost and benefits of minor changes in group size, but also in ASR, are more pronounced, compared to groups that are—say—three or four times larger[47]. Because changes in the ASR of Verreaux's sifakas may be accompanied by fundamental changes in the mating system (single- vs. multimale group) and because the magnitude of ASR variation at small group sizes is uncorrelated with the number of adult females, these factors may have independent effects on the optimal group size. Finally, our study is of particular interest because a previous report on the same population revealed that intergroup variation in several commonly used fitness proxies, such as daily travel distance, home range size, foraging rate, resting rate, faecal glucocorticoid metabolites concentration and parasite richness were all uncorrelated with group size[53]. Here, we use two direct fitness measures—birth rates and survival—obtained during a 25-year field study, and predicted either no effect of group size, if our earlier used indirect fitness proxies are functionally linked to reproduction and survival, or higher reproductive and survival rates in groups of intermediate size, as predicted by the optimal group size hypothesis[44–46].

## Results
### Group size variation
Group size varied between 2 and 10 individuals (mean: 6.26, SD = 1.96; Fig.1). The average number of adult females, adult males and juveniles were 1.93 ± 0.80, 2.13 ± 0.86, and 2.20 ± 1.34, respectively.

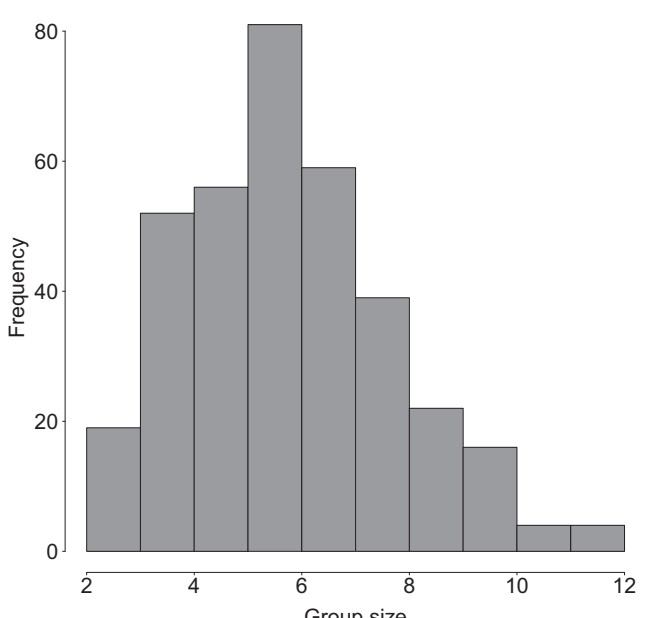

**Fig. 1 | Group size variation in Verreaux's sifaka at Kirindy Forest between 1994 and 2020.** Depicted is the frequency distribution of group sizes in April of each year (N = 352).

### Reproductive success
The probability of giving birth was best predicted by the model including linear terms (AIC: linear terms = 425.99, quadratic terms = 434.96). The probability of giving birth was not significantly affected by the size of the group a female lived in (likelihood ratio test full-null model comparison: $X^2$ = 29.26, df = 3, p < 0.001; Fig. 2a; Table S1a). However, females in groups with more adult females were less likely to give birth (Fig. 2b). Moreover, female age had a significant effect on the probability of giving birth, with older females exhibiting the lowest birth rates (Fig. 3). Year-to-year variation in resource abundance, as indexed by cumulative rainfall in the year before giving birth had no effect on the probability of giving birth. The model including inter-group variation in ASR revealed similar effects, but ASR was not significant (likelihood ratio test full-null model comparison: $X^2$ = 22.18, df = 5, p < 0.001; Table S1b). Since AICs of the model including linear or quadratic terms did not differ by much (AIC: linear terms = 441.01, quadratic terms = 440.17), we present here the more comprehensive model including quadratic terms. Finally, birth rates were not impeded by dominance status (Table S2c-f).

### Survival
Annual survival was best predicted by the model including the quadratic terms (AIC: linear terms = 818.68, quadratic terms = 764.28). Annual survival was positively affected by group size, with individuals in larger groups experiencing a significantly higher mortality risk (likelihood ratio test full-null model comparison: $X^2$ = 55.02, df = 5, p < 0.001; Fig. 4a; Table S1c). Age predicted the probability of dying as well, with younger individuals exhibiting higher mortality (Fig. 4b). Neither ASR nor rainfall in the year preceding a death were significant predictors of mortality risk, however (Table S1c). Models including only confirmed deaths or confirmed deaths as well as all disappeared females revealed similar effects (Table S3).

## Discussion
There is apparently no intermediate optimal group size in Verreaux's sifakas at which both reproduction and survival are maximised because the corresponding costs and benefits of group size variation are distributed asymmetrically. While sifakas enjoyed higher survival in smaller groups and higher birth rates in groups with few other adult females, there are apparently no net fitness benefits of living in large groups, creating overall selective

**Fig. 2 | Group size effects on fitness proxies in Verreaux's sifaka at Kirindy Forest between 1994 and 2020. a** Probability of giving birth as a function of group size and (**b**) female age (*N* = 352). Size of circles represents number of events (**a**) range = 3–81, (**b**) range=10-68. Dotted line: regression line; shaded area: 95% confidence intervals.

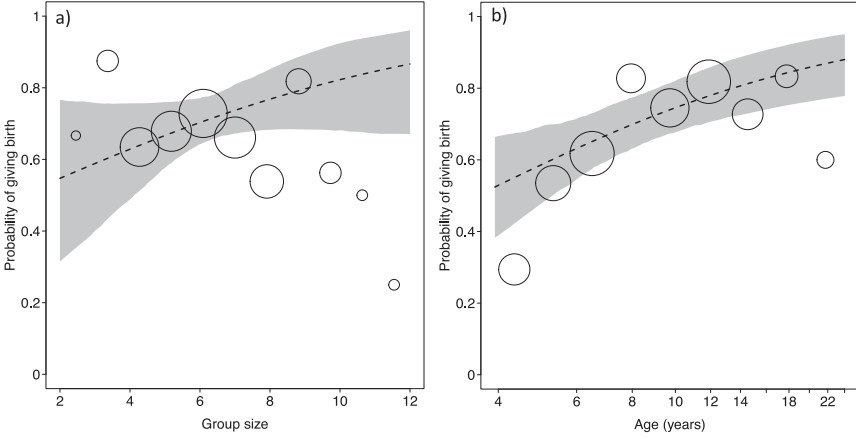

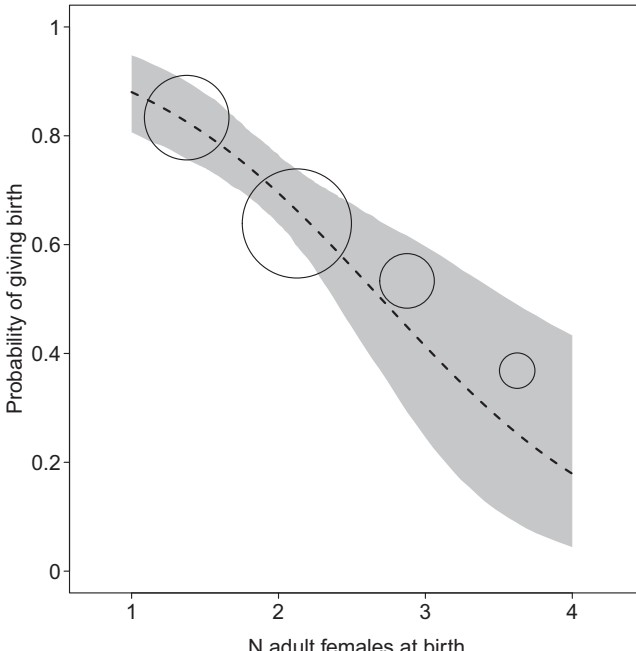

**Fig. 3 | Reproductive inhibition among female Verreaux's sifaka at Kirindy Forest between 1994 and 2020.** The number of co-resident adult females has a significant negative effect on the probability of giving birth (*N* = 352). Size of circles represents number of events. Dotted line: regression line; shaded area: 95% confidence intervals.

pressures favouring small groups. Our study also revealed that the different fitness measures exhibit different relationships with overall group size, indicating that lifetime reproductive success (of females) is probably not described by a simple linear function of overall group size and presumably more closely related to certain components of group size, e.g., the number of females competing over reproductive opportunities. Hence, different fitness components can vary independently across group sizes. Furthermore, our previous study of multiple indirect fitness proxies[53], which are more commonly recorded in other studies as well, has presumably not missed any major effects, but they may be individually too weak to predict the survival costs of large group sizes. Finally, despite living in a highly seasonal environment with pronounced year-to-year variation in the length and intensity of the wet season, environmental variation did not have an effect on either fitness proxy. We discuss these major findings in detail below.

First, overall group size had no effect on the probability that a female sifaka will give birth. Like virtually all other lemur species, Verreaux's sifakas are seasonal breeders that give birth in the middle of the local ca. 8-month

dry season, so that infant weaning coincides with the season of greatest food availability at the end of the wet season[54]. On average, not every female gives birth every year, but this intra- and interindividual variation in female reproductive success is not explained by group size. Furthermore, variation in group composition, which we quantified as ASR, had no effect on birth rates, suggesting that females do apparently not enjoy any of the potential benefits in the context of reproduction that can accrue as a result of male services in other primates[55,56].

Instead, an aspect of group size defined by a subset of individuals—the number of co-resident adult females—best predicts variation in birth rates. This result indicates that female Verreaux's sifakas are subject to subtle forms of female competition over reproduction because the probability of giving birth declined with increasing numbers of adult females. Female Verreaux's sifakas are for the most part philopatric[57]. Only a small proportion of them were observed to transfer between groups; typically, if a group had three or more adult females and a breeding vacancy in an adjacent group appeared, e.g. after the death of the only breeding female[58]. Thus, in the vast majority of cases, co-resident adult females are close relatives. They exhibit clear dominance relationships[59], but dominants do not appear to actively suppress reproduction in subordinates. Hence, competition over reproduction affects all females, but we do not yet know whether the underlying mechanism is based on behaviour and or physiology. In sympatric redfronted lemurs and other members of the Lemuridae, as well as in meerkats and banded mongooses, females target co-resident females for temporary or permanent eviction[52,60–62], induce abortions[63] or commit infanticide[64]. Thus, in comparison, Verreaux's sifaka females engage in relatively subtle forms of reproductive competition.

What do female Verreaux's sifakas compete for? Generally accepted benefits of small group size include a reduction in the intensity of within-group feeding competition and reduced travel costs in smaller home ranges (assuming homogeneity in habitat quality[53]), both of which should make more energy available for reproduction. By compromising reproduction in group mates, females may contribute to a limitation of group size in the near future because juvenile sifakas develop and reach nutritional independence much faster than other primates of their size[65] and would compete for food with adults already in their first year of life. Additional energy for reproduction cannot be used to increase fecundity because litter size is invariably equal to one[51]. A future analysis should therefore examine whether age of first reproduction occurs earlier, average inter-birth intervals are shorter and infant survival is higher in smaller groups; a pattern previously described for Phayre's leaf monkeys, where infants in larger groups developed more slowly, were weaned later and inter-birth intervals were larger[36].

The nature and intensity of food competition may also depend on climatic variation and competition among neighbouring groups. However, group size does only play a subordinate role in predicting success in inter-group encounters[66] because only proximity to the core area influenced the probability of winning[67]. Inter-annual variation in rainfall in Western

**Fig. 4 | Age effects on fitness proxies in Verreaux's sifaka at Kirindy Forest between 1994 and 2020.**
**a** Probability of dying as a function of group size and (**b**) age (N = 1022). Size of circles represents number of events (**a**) range = 21–354, (**b**) range = 7–371. Dotted line: regression line; shaded area: 95% confidence intervals.

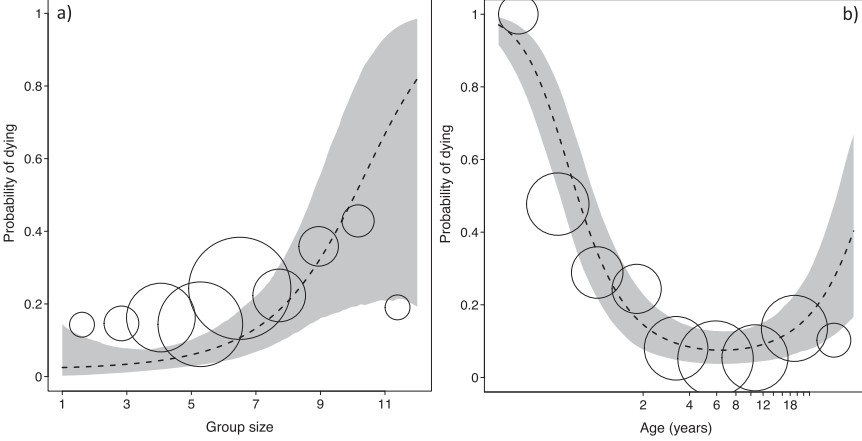

Madagascar is substantial in the onset, duration and total yield during the 3–4 months long wet season. Mating in sifakas takes place in February near the end of the wet season, when better female condition in a much smaller sample predicted the probability of giving birth 5 months later[54,68]. Assuming that rainfall, which has decreased substantially over the last decades[50], is positively correlated with food availability and, ultimately, female condition, we would have expected to find a positive effect on birth rates as well. However, these energetic considerations are apparently much less important than some intrinsic (age) and social (number of co-resident females) factors.

It is finally also possible that group composition has a stronger effect on reproductive success than group size. In contrast to other studies (e.g. ref. 38), we therefore included males in our analyses of group size. While variation in ASR had no effect on reproductive rates, there should be sexual conflict over group composition with down-stream effects on group size. Male Verreaux's sifakas also exhibit clear dominance relations, including phenotypic modification of the highest-ranking male[69] and physiological suppression of subordinates[70]. They exhibit one of the highest levels of reproductive skew among mammals, with the highest-ranking male acquiring on average >90% of paternities in a group[71]. Male monopolisation of oestrous females is achieved by mate guarding and facilitated by oestrus asynchrony[72]. Dominant males should therefore benefit reproductively from an increase in the number of adult females[73], revealing an apparent sexual conflict with females over group composition. Similarly, the share of paternities of subordinate males increases with the number of females[74,75], aligning the reproductive interests of dominant and subordinate males. Thus, the groups of Verreaux's sifakas with above-average size may reflect the effects of male reproductive interests. However, because sifaka females, as those of many other lemur species[76], unconditionally dominate males[77], this sex difference in agonistic power may proximately explain why females win this sexual conflict on average, and ultimately explain the—compared to anthropoid primates—relatively small group sizes of this and other lemur species[78]. However, reports of male infanticide[79] suggest that females are not always able to prevent the immigration of males. In horses and zebras, there may be a similar conflict of interest over the optimal group size among females, subordinate males and dominant males that is resolved by female dispersal[80]. Thus, the costs and benefits of a given group size appear to be shaped by more factors than the number of individuals as well as by the behavioural mechanisms available to different classes of individuals to resolve the arising conflicts.

Furthermore, reproductive success was influenced by age, with young and old females exhibiting the lowest birth rates. In a recent study on reproductive senescence in several lemur species, including Verreaux's sifaka, we analysed variation in the probability of giving birth within and across females as a function of their age[51]. In contrast to this previous study, the present analyses include an additional year of data, and, hence, more births and mothers. Furthermore, in the present analyses, we included females with an age of 4 years and older because that is the youngest age at which females have ever been observed to give birth. In the previous analyses, we only included females after they had given birth for the first time. This different classification should explain why we found an effect of age on the probability of giving birth in the present analysis, but not in ref. 51.

Second, the probability of dying was best predicted by group size, but in the opposite direction from what conventional wisdom holds. Focusing on potential causes of extrinsic mortality, several simple mechanisms, like the dilution or predator confusion effect[1], are thought to generate survival benefits for individual living in larger groups because of reduced predation risk, which, in turn, should result in greater longevity. However, these advantages were neither evident in comparative analyses across more than 250 species of mammals[81], nor across more than 400 species of birds[82]. It is therefore possible that larger groups may be actually more conspicuous for predators or that more eyes and ears cannot defend against certain types of predators that hunt by immediate attacks or ambush[83]. Large group size may affect survival also indirectly because individuals experience greater feeding competition that may translate into greater variance in body condition[84].

Moreover, intrinsic causes of mortality may exacerbate the costs of living in large(r) groups and contribute to the observed effect in Verreaux's sifaka and perhaps more generally. In general, the risk of parasite transmission among group mates indeed increased weakly with group size[85], but it can also be mitigated by social network metrics[86] and properties[87]. Moreover, physiological costs of living in larger groups, such as increased energy expenditure and feeding competition, are not linearly elevated in larger groups[88]. Various proxies of these costs were also not consistently related to group size in Verreaux's sifaka[53], and higher group size provided no advantages in intergroup competition in access to resources[66,67]. At other long-term study sites of Verreaux's sifakas to the south of Kirindy Forest, where seasonality is more pronounced and annual rainfall reduced, and where key predators like the fosa are less abundant or totally absent[89,90], the average group size is also at around 6 individuals (Berenty[91]; Hazafotsy[92]; Beza Mahafaly[57,93]), indicating that group size is rather resilient to massive variation in key environmental factors.

In conclusion, patterns of fitness consequences of group size variation in Verreaux's sifakas do not follow theoretical predictions, indicating that the optimal group size model may be oversimplistic. More specific aspects of group size, such as the number of adult females in this study, may be additionally important drivers of fitness consequences beyond group size per se. Similarly, sexual conflict over group composition can also have repercussions for group size that deserve more scrutiny. Also, survival and reproductive success seem to vary more often than not independently of each other as a function of group size; i.e., different fitness components are evaluated independently. Hence, there is a need for additional, more comprehensive and fine-grained studies to unravel the drivers of intraspecific variation in group size in order to finally clarify one of the oldest questions in behavioural ecology.

## Methods

### Study site and species

This study was conducted on a population of Verreaux's sifakas at Kirindy Forest, Western Madagascar. Kirindy Forest is a protected dry deciduous forest and subject to pronounced seasonality, with a long, cool dry season (April to October) and a hot wet season (November to March), harbouring a full set of local sifaka predators, including fosas and Harrier hawks[58]. Verreaux's sifakas are diurnal and arboreal primates with a frugi-folivorous diet[94]. They can live up to 25 years (median longevity: 12 years) and can produce a single infant every year (mean interbirth interval: 1.25 years) from their fourth year of life (mean age of first reproduction: 5.6 years) onwards without any sign of reproductive senescence[51]. Since 1994, all animals within a ca. 60 ha study area have been individually marked with unique nylon or radio collars, either when they were about 8 months old or when they immigrated into one of the study groups[53]. Here, we include data from a total of 279 Verreaux's sifakas living in up to 10 adjacent groups.

Our analyses are based on demographic data collected during multiple weekly censuses of all study groups. Between 1994 and 2021, we recorded a total of 236 births that were distributed among 41 mothers. Female lifetime reproductive success ranged from 1 to 16 infants (mean: 5.75 infants). We classified a total of 105 individuals (18 females, 21 males, 66 of unknown sex) as dead. An individual was pronounced dead when either a successful predation event was observed, its remains were found or when the individual was less than 8 months old at the time of disappearance, and, hence, barely weaned[65]. An additional total of 162 individuals (66 females, 89 males, 7 of unknown sex) disappeared from their group. They were classified as "unknown", as we could not unequivocally establish whether they had emigrated from the study area or died. Twenty individuals (7 females, 13 males) were alive at the time of data acquisition for this study.

### Statistics and reproducibility

We constructed binomial General Linear Mixed Models with a logit link function to estimate factors predicting the likelihood of both, giving birth to an offspring or dying. We included group size (i.e., the number of all members of a group, excluding dependent infants at the time of a birth or death, respectively), the age of the female at the time of giving birth or of the dead individual at the time of its death (in years), the ASR, (i.e., the proportion of adult males among the adults of a group[95], with individuals with ≥5 of age being counted as adult[58]), and the total amount of rainfall at the study site in the 12 months prior to the birth season in July, using rainfall estimates detailed in ref. 50. Because group size and age may have non-linear effects[51], we also tested the effects of quadratic group size and quadratic age. The model on birth rates also included the number of co-resident adult females and the quadratic term of it as a factor because preliminary analyses indicated a potentially inhibitory effect[58]. We compared models including the linear and quadratic effects by using Akaike information criterion (AIC) and present the models with a delta AIC < 2 in the main text and all other models in the Supplementary Information (Table S2) We did not include higher order polynomials because we are not aware of any hypothesis that the probability of giving birth or annual survival should follow a cubic or higher order function. Because the number of adult females and ASR were collinear, we estimated two models for birth rates; one including the number of adult females and one including ASR.

For the birth rate model including the number of adult females, we set the occurrence of whether a female gave birth (yes, no) as response and included group size and quadratic group size, females age and quadratic age, the number of adult females and quadratic number of adult females, as well as annual rainfall as fixed factors. We included individual and group identity as random factors, with random slopes of group size and quadratic group size, females age and quadratic age, the number of adult females and quadratic number of adult females, and rainfall within individual and group identity, respectively. Originally, we also included correlations between random slopes and intercepts, but as the models did not converge, we excluded them again. For the birth rate model including ASR, we fitted the same model but included ASR instead of the number of females.

In addition, to estimate the effect of dominance status on birth rates, we fitted two more models on a reduced data sets comprising only groups that contained at least two adult females (Table S2). In these models, we included the same factors as above and dominance status, i.e., whether a female was dominant or not.

To estimate annual survival, we fitted three models including different death classifications: a) only confirmed dead individuals ($N = 105$), b) confirmed dead combined with "unknown" females because females only leave their natal group in rare, exceptional circumstances ($N = 173$, 83 females, 21 males, 67 of unknown sex)[58], and c) confirmed dead, "unknown" females and males that were ≤4 of age because the youngest recorded emigrated male was 5 years old ($N = 216$, 83 females, 65 males, 67 of unknown sex)[58]. We set the occurrence of whether an individual was classified as dead (yes, no) as response and included group size and quadratic group size, individuals' age and quadratic age, ASR, and annual rainfall as fixed factors. We included individual and group identity as random factors. For model a) we originally included random slopes of group size and quadratic group size, individuals' age and quadratic age, ASR, and annual rainfall within groups without correlations between random slopes and intercepts but removed the age terms as the model did not converge (Table S1c). For model b) and c) we included all random slopes without correlations between random slopes and intercepts. Since the results of models a and b revealed similar effects, we present only model c), which is based on the largest sample size, in the main text. Results of models a) and b) are reported in the Supplementary Information (Table S3).

### Model implementation

All analyses were conducted using R (version 4.3.2, R Core Team 2023). To ease model convergence, we log-transformed age and rainfall, and centred all predictors to a mean of zero and a standard deviation of one before including them into the models. We included all theoretically identifiable random slopes to avoid Type I errors[96]. To test the significance of predictors as a whole, we compared the fit of the full model with that of the null model comprising only random factors[97,98]. We obtained confidence intervals for all models by means of parametric bootstraps using the function "bootMer" of the package "lme4", applying 1000 parametric bootstraps. We checked for collinearity by determining Variance Inflation Factors (VIF) for a standard linear model without random effects using the package "car" (version 3.0.11[99]). To estimate model stability, we proceeded by dropping levels of the random effect one at a time from the data set and compared the obtained estimates to the estimates obtained for the full data set. All models revealed good model stability.

### Reporting summary

Further information on research design is available in the Nature Portfolio Reporting Summary linked to this article.

### Data availability

Data and code are provided at https://figshare.com/s/c4fc966c04e6b36de6ea

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

## Acknowledgements

We thank the Malagasy Ministère de l'Environnement, the members of the CAFF/CORE, the Mention Zoologie et Biodiversité Animale de l'Université d'Antananarivo and the Centre National de Formation, d'Etudes et de Recherche en Environnementet Foresterie de Morondava for authorising and supporting our research in Kirindy Forest. We are grateful to Tiana Andrianjanahary, Mamy Razafindrasamba and other field assistants for their support in data collection. This work was supported by the German Science Foundation, DFG [grant numbers Ka 1082/9-1&2, Ka 1082/29-2] awarded to P.M.K. We are grateful to Severine Hex and an anonymous referee for constructive comments on this paper.

## Author contributions

P.K. and C.F. designed the study. C.F. analysed data. P.K. wrote the manuscript and C.F. critically revised it.

## Competing interests

The authors declare no competing interests.

## Ethics

We adhered to the "Guidelines for the ethical treatment of nonhuman animals in behavioural research and teaching" as published in Animal Behavior (2023, 195, I-XI) and the laws of the country where the research was conducted. This study was approved by the Département de Biologie Animale of the University of Antananarivo and the CAFF/CORE of the Direction des Eaux and Forêts de Madagascar.
