## [Peer Review File · Communications Biology]

Reviewers' comments:

Reviewer #1 (Remarks to the Author):

This is an interesting paper based on an outstanding dataset covering reproductive success and survival in different sized groups of sifakas. It has a rather strange structure and would be improved by a clearer description and justification of the questions that it is asking in the Introduction. In particular, it might also be useful to make it clear at the outset that variation in group size is a product of variation in rates of reproduction, survival, immigration and emigration in different age and sex categories; that group size should be expected to have different effects on the fitness of different age/sex/status categories of individuals; and that their ability to control group size varies. As a result, should we really expect variation in group size to reflect the optimum group size for all group members? Might it not be better to set this up as a prediction in the Introduction and then to test it rather than reaching this conclusion by a circuitous route? In addition, the Discussion includes a mixture of review information from comparative studies and discussion of the sifaka results. Might it not be preferable to place the review of comparative results in the Introduction and use them to set up the questions to be tested – and to focus the Discussion at first on the sifaka results and then subsequently on general trends?

DETAILED COMMENTS

Page 1

While it has often been suggested that the evolutionary mechanisms responsible for interspecific and intraspecific variation in group size may be similar, this isn't necessarily the case and quite different processes may be responsible for inter- and intra-specific variation in sociality.

Page 2; para 2

It is commonly accepted that group size has multiple benefits and that these vary between species – so is it surprising that relationships between group size and fitness differ between species?

Page 2; para 3

Was it really the variation in relationships between group size and fitness that generated the practice of controlling for other parameters? Wasn't this an inevitable consequence of the realisation that many different aspects of individuals affected the breeding success of individuals and that measuring the effects of group size make it necessary.

Page 2; para 4

Thus? Why does evidence of interspecific variation in group size/fitness relationships suggest that there should be stabilising selection?

Page 2; para 4

Selection for different group sizes will vary between age, sex and dominance categories – as will the capacity to control it. Arguments that selection will necessarily favour intermediate group sizes are dubious.

Page 3

Group size is a consequence of variation in rates of breeding, survival, immigration and emigration. In some species particular age categories control one or more of these parameters while in others the control may be more widely distributed or be lacking altogether. To understand variation in group size it is consequently important to understand the relative roles of reproduction, survival, immigration and emigration – and the extent to which individuals control them.

Page 7; para 1

Don't you need to consider the likely persistence of small and large groups – which is likely to affect

the inclusive fitness of females.

Page 7; para 1

Can you measure the relationship between group size and the lifetime reproductive success of females? If so, do present it.

Page 7; para 2

Different fitness components can vary independently across group sizes. Is this really novel? Hasn't it been shown in other mammals? Lions? Hyenas?

Page 8; para 1

Is there any effect of group size on reproduction in any grouping of females (eg dominants)?

Page 8; para 2

This is surely what would be expected - that the numbers of particular age/sex categories would often be most likely to affect the fitness of the same group? There will, however, be exceptions eg the results of effects of male harassment may sometimes indicate that male numbers affect females.

Page 10; para 2

Or that larger groups generate greater competition for resources and greater variation in weight and condition?

Apart from suggestions for tightening the structure of the paper, my principal comment would be that, if data are available, it would be useful to examine the relationship between group size and the lifetime breeding success of females and the persistence of particular matriline and groups.

Reviewer #2 (Remarks to the Author):

General comments:

This paper utilized a longitudinal dataset spanning 25 years to investigate the fitness consequences of group size in female Verreaux's sifaka. The authors conclude that the number of co-resident females, but not overall group size, suppresses reproduction in females, while mortality is higher in groups of larger size. Taken together, the authors suggest that independent fitness measures suggest that optimal group size is small in this system due to there being no benefit of large group size and no consequences of small group size. This paper therefore contributes an important degree of nuance to the body of work investigating the forces which govern group size in social organisms.

Overall, I find this study to be well presented, straightforward, and a valuable contribution to the field. My primary recommendations are for a bit more justification, species-specific context, and more explicitly articulated predictions in the introduction to frame the remainder of the paper. A few places could further benefit from editing for clarity and grammar to hone the author's message.

Introduction:

Page 1: remove comma- "Similar selection pressures are thought to govern both interspecific evolutionary transitions in social organization and intraspecific variation in group size."

Page 2, paragraph 2: "Second, variation in social organization, and group size in particular, is associated with

much more variation in various fitness measures, but these effects vary within and across species." – I would recommend re-wording this sentence. The word variation/various/vary occurs 4 times within the sentence, and the meaning is a bit opaque. It seems the authors are trying to convey that variation in social organization is associated with a wider array of fitness measures, though the effect is heterogenous across and within species?

I also recommend organizing these examples in terms of those which showed similar findings to

highlight the contradictions across species and aid in the reader following the diversity of findings.

Page 2, 3rd paragraph: you introduce the idea of a presumed intermediate optimum, but aside from the preceding example of marmots, there is not much discussion of why an intermediate group size should be optimal. This seems important to emphasize, as in your discussion you assert that there is no cost to small group size, but no benefit to large group size, meaning the optimum is small rather than intermediate. I would appreciate perhaps some discussion of the potential costs of small groups that would be acting against the potential benefits of larger groups in addition to the apparent costs of very large groups that you provide.

Page 2, final paragraph: This paragraph begins with "thus, in order to optimize..." I would appreciate a bit more justification for this statement, as not all of your preceding examples lend themselves obviously to the conclusion that stabilizing selection on intermediate group sizes is to be expected (certainly the case in the yellow-bellied marmots and the anis, for instance, but it is less clear for the giraffe example). Perhaps this would simply require a bit of reordering between the preceding paragraph and this one to aid in the flow of logic from the discussion of fluctuating benefits across different environmental conditions to the expectation of an intermediate optimum.

Page 3, paragraph 2: You mention including ASR and interannual variation in rainfall to put the present study in the context of other studies. This is understandable, but I would appreciate a little discussion here on why these variables might be considered biologically relevant to the present study species and whether the authors had any predictions as to how these factors might influence fitness outcomes in female sifakas.

I also would recommend mentioning the species in question in the sentence "Furthermore, because a previous study suggested that female competition might impact reproductive rates in a sympatric species". Further, are there any other similarities in social system between Verraueux's sifakas and redfronted lemurs (i.e. female sympatry?) which would suggest a similar pressure acting in both species? If so, I would mention that, because at present it seems as though the only justification was that a sympatric species showed this pattern, which does not seem quite a sufficient justification on its own.

Would the authors be able to clarify why the number of adult females vs. the ASR might yield different results, as was observed in your study?

Discussion

Page 7: "While sifakas enjoyed higher survival in smaller groups and higher birth rates in groups with few other adult females, there are apparently no net fitness costs of living in small groups and no net fitness benefits of living in large groups, creating overall selective pressures favouring small groups" – this sentence makes it seem as though the second half is somehow in contradiction with the findings in the first clause, which do not appear to be the case. Both of the benefits presented in the first clause (higher survival in small groups and higher birth rates in groups with few other adult females, which also would be smaller groups by definition) suggest that all directionality of selection is going towards small groups, as you state. Therefore, a small rewarding might be beneficial.

Page nine, paragraph 2: "Group size does only play a subordinate role in predicting success in inter-group encounters 60, however, because proximity to their core area, rather than the numerical advantage of a group in a given encounter influenced the probability of winning" – I suggest rewording this sentence for clarity.

Same paragraph: I think some of this justification for including intra-annual variation in rainfall and food availability could be relocated to the introduction.

Page nine, final paragraph: missing a closing parentheses around the citation.

"Dominant males should therefore benefit reproductively from an increase in the number of adult females 67, revealing an apparent sexual conflict with females over group composition" – however, might there not be another side to this, in which males might be expected to benefit up to a certain

point, after which they would not be able to monopolize reproduction as effectively as they would like? In feral horses and zebras, which also have relatively small group sizes, males might be predicted to want larger harems only up until they are no longer able to defend paternity (e.g. Rubenstein & Nunez, 2009). I bring up equids because at least in terms of how much paternity a dominant male can monopolize and the proximate mechanisms that facilitate it (e.g. estrus asynchrony) appear to be similar, and group sizes also appear to be similar on average. The authors suggest that there would be an alignment between the dominant and subordinate male's interests, but I would argue that there might actually be three independent optima, in which females might prefer the smallest groups, subordinates might prefer the largest, and dominant males might prefer group sizes intermediate but perhaps closer to that of females than that of subordinate males. If this is the case, there could be a greater alignment between dominant males and females. I raise this as a potential alternative, and not necessarily mutually exclusive hypothesis for the author's consideration alongside the proposed mechanism.

The authors suggests that there is a male and female conflict of interest and that female dominance wins in this intersexual conflict. I am curious if there is there any evidence to suggest that this conflict may be playing out in the form of males attempting to recruit or retain females and females attempting to prevent immigration or induce dispersal/eviction?

Page 10, paragraph 1: the authors mention more births and mothers were included in the present study compared with the previous one. How much larger was the sample size in the present study once these classification changes were implemented?

Page 11, paragraph 1: "Also, survival and reproductive success seem to vary more often than not independently as a function of group size; i.e., different fitness components are evaluated independently." – I would suggest rewording for greater clarity.

Methods

Statistical and data collection methods appeared appropriate to the study aims.

Reviewers' comments:

Reviewer #1 (Remarks to the Author):

This is an interesting paper based on an outstanding dataset covering reproductive success and survival in different sized groups of sifakas. It has a rather strange structure and would be improved by a clearer description and justification of the questions that it is asking in the Introduction. In particular, it might also be useful to make it clear at the outset that variation in group size is a product of variation in rates of reproduction, survival, immigration and emigration in different age and sex categories; that group size should be expected to have different effects on the fitness of different age/sex/status categories of individuals; and that their ability to control group size varies.

This is a constructive suggestion that we happily integrated. We made the causes of group size variation explicit and added dominance to the effects of sex and age already mentioned in the last sentence of the first paragraph of the original ms.

As a result, should we really expect variation in group size to reflect the optimum group size for all group members?

However, we beg to differ from the assertion that the structure is strange and that the questions that we ask are not/poorly justified for two main reasons. First, as the Introduction and the >30 studies cited therein show, we are relying on the traditional theoretical framework that has guided studies on the optimal group size for decades. Our summary of the literature of the relevant studies on mammals indicated that this body of research has used different fitness proxies and produced divergent results. Studies of additional species are therefore "justified", especially since we are one of the very few studies to examine **two** fitness **determinants** (and not just proxies like feeding or vigilance rates) simultaneously.

Second, while it is obvious that individuals of different age/sex classes should have different group size optima, we are looking at the results of their integration at the group level. To our knowledge, no one has accepted the challenges of determining the group sizes that maximize survival for juvenile and adult males and females, as well as those that maximize reproductive success of adult females and males (which would require genetic testing) **separately**. Should there be interactions between the interests or effects of members of different age-sex classes, such a massive effort may not even produce meaningful results.

Might it not be better to set this up as a prediction in the Introduction and then to test it rather than reaching this conclusion by a circuitous route?

We think the referee refers here to our specific result that the number of adult females is a better predictor of birth rates than total group size (but not mortality!?!). This is a new result that was neither predicted by theory nor by results of previous studies. It is therefore not possible to set this prediction up in the Introduction based on any prior information.

In addition, the Discussion includes a mixture of review information from comparative studies and discussion of the sifaka results. Might it not be preferable to place the review of comparative results in the Introduction and use them to set up the questions to be tested - and to focus the Discussion at first on the sifaka results and then subsequently on general trends?

In our Discussion, we discuss each major finding also in a comparative perspective in one coherent paragraph because the suggested alternative (to discuss sifaka results and general trends separately) would introduce too much redundancy and less coherence.

DETAILED COMMENTS

Page 1

While it has often been suggested that the evolutionary mechanisms responsible for interspecific and intraspecific variation in group size may be similar, this isn't necessarily the case and quite different processes may be responsible for inter- and intra-specific variation in sociality.

We appreciate this cautionary note and added a sentence to the only relevant study familiar to us that emphasizes this caveat (Gygax 2002).

Page 2; para 2

It is commonly accepted that group size has multiple benefits and that these vary between species - so is it surprising that relationships between group size and fitness differ between species?

The points that we make here is that i) (most) previous studies used a single fitness proxy, ii) even though different proxies of survival and reproductive success were used, different studies (sometimes of the same species) revealed different patterns (positive, negative or no correlation with group size), iii) previous studies of primates yielded a somewhat more consistent pattern than studies of other taxa. Highlighting this general pattern is important because it contrasts with fundamental predictions of the corresponding theory about an optimal intermediate group size. We added a sentence to clarify this aspect.

Page 2; para 3

Was it really the variation in relationships between group size and fitness that generated the practice of controlling for other parameters? Wasn't this an inevitable consequence of the realisation that many different aspects of individuals affected the breeding success of individuals and that measuring the effects of group size make it necessary.

We do not know what motivated the authors of this set of studies, but we deleted the part of the opening sentence that created this impression to produce a more neutral opening sentence to this paragraph.

Page 2; para 4

Thus? Why does evidence of interspecific variation in group size/fitness relationships suggest that there should be stabilising selection?

Here, we highlight the mismatch between theoretical predictions about an optimal group size (which implies stabilizing selection) and the state of the empirical literature summarized above to motivate the significance of additional studies (like ours) to contribute additional data and insights. We think that the reference to three studies that detail these predictions (see e.g. Markham et al. (2015) PNAS) is sufficient to underscore this point.

Page 2; para 4

Selection for different group sizes will vary between age, sex and dominance categories - as will the capacity to control it. Arguments that selection will necessarily favour intermediate group sizes are dubious.

This is an excellent relevant additional point. We integrated it into this paragraph. Given a potential diversity of outcomes for different group sizes and for different age-sex classes, we think it is neither dubious nor surprising that they converge upon an average intermediate value.

Page 3

Group size is a consequence of variation in rates of breeding, survival, immigration and emigration. In some species particular age categories control one or more of these

parameters while in others the control may be more widely distributed or be lacking altogether. To understand variation in group size it is consequently important to understand the relative roles of reproduction, survival, immigration and emigration – and the extent to which individuals control them.

Same here. We added a sentence with this general statement to better explain our analytical strategy.

Page 7; para 1

Don't you need to consider the likely persistence of small and large groups – which is likely to affect the inclusive fitness of females.

It is only possible to link the fitness-relevant events (births and deaths) to the group size at that moment in time when they occur. Because group size is a dynamic variable that changes with every birth and death, but also every immigration or emigration, it is not possible to link these events to the persistence of a particular group size over years or decades.

Page 7; para 1

Can you measure the relationship between group size and the lifetime reproductive success of females? If so, do present it.

Here, we have the same problem. Over a lifetime of say 20 years, the group size a given female may find itself in has typically changed many times. As a result, we cannot link individual births and deaths to a "lifetime group size". Instead, we use the instantaneous group size for each event and average that across groups and time.

Page 7; para 2

Different fitness components can vary independently across group sizes. Is this really novel? Hasn't it been shown in other mammals? Lions? Hyenas?

The results of a careful and extensive literature search for the relationship between group size and fitness in mammals are summarized on p. 2 (2nd para) of the original ms. We are not aware of any relevant studies in lions (where only hunting success and dispersal were studied as a function of group size) or hyenas (group size was only included as a covariate in one study of one clan, which we have added to the relevant paragraph on p. 2).

Page 8; para 1

Is there any effect of group size on reproduction in any grouping of females (eg dominants)?

This is a good question, but we did not address it for two reasons. First, about half of the groups included only one adult female during many years of the study. In that situation, it is impossible to determine dominance relations and to examine their potential effects. Second, for those group constellations with 2 or more females, we do not always have behavioral data that would allow such a classification. Furthermore, we already discussed this topic in the subsequent paragraph in the original manuscript.

Page 8; para 2

This is surely what would be expected - that the numbers of particular age/sex categories would often be most likely to affect the fitness of the same group? There will, however, be exceptions eg the results of effects of male harassment may sometimes indicate that male numbers affect females.

The fact that the number of adult females would be a better predictor of birth rates than total group size was neither predicted nor expected because neither theory nor previous studies of other species indicated the existence of such an effect. Since we did not frame this insight as "unexpected", we did not change our wording here.

Page 10; para 2

Or that larger groups generate greater competition for resources and greater variation in weight and condition?

That's a good point. We added a sentence about this factor along with a reference to a recent study that appears to support this mechanism.

Apart from suggestions for tightening the structure of the paper, my principal comment would be that, if data are available, it would be useful to examine the relationship between group size and the lifetime breeding success of females and the persistence of particular matriline and groups.

As explained above, group size is too variable and life expectancy is too long to generate any meaningful measure of "lifetime group size".

Reviewer #2 (Remarks to the Author):

General comments:

This paper utilized a longitudinal dataset spanning 25 years to investigate the fitness consequences of group size in female Verreaux's sifaka. The authors conclude that the number of co-resident females, but not overall group size, suppresses reproduction in females, while mortality is higher in groups of larger size. Taken together, the authors suggest that independent fitness measures suggest that optimal group size is small in this

system due to there being no benefit of large group size and no consequences of small group size. This paper therefore contributes an important degree of nuance to the body of work investigating the forces which govern group size in social organisms.

Overall, I find this study to be well presented, straightforward, and a valuable contribution to the field.

Thank you for this overall very positive assessment.

My primary recommendations are for a bit more justification, species-specific context, and more explicitly articulated predictions in the introduction to frame the remainder of the paper. A few places could further benefit from editing for clarity and grammar to hone the author's message.

Introduction:

Page 1: remove comma- "Similar selection pressures are thought to govern both interspecific evolutionary transitions in social organization and intraspecific variation in group size."

Done.

Page 2, paragraph 2: "Second, variation in social organization, and group size in particular, is associated with much more variation in various fitness measures, but these effects vary within and across species." - I would recommend re-wording this sentence. The word variation/various/vary occurs 4 times within the sentence, and the meaning is a bit opaque. It seems the authors are trying to convey that variation in social organization is associated with a wider array of fitness measures, though the effect is heterogenous across and within species?

Done.

I also recommend organizing these examples in terms of those which showed similar findings to highlight the contradictions across species and aid in the reader following the diversity of findings.

We appreciate this proposal and its advantages. However, our summary of the literature indicated a possible taxonomic bias. We therefore organized this information by taxonomic order (carnivorans, rodents, primates) to highlight this apparent effect.

Page 2, 3rd paragraph: you introduce the idea of a presumed intermediate optimum, but aside from the preceding example of marmots, there is not much discussion of why an intermediate group size should be optimal. This seems important to emphasize, as in your discussion you assert that there is no cost to small group size, but no benefit to large group size, meaning the optimum is small rather than intermediate. I would appreciate perhaps some discussion of the potential costs of small groups that would be acting against the potential benefits of larger groups in addition to the apparent costs of very large groups that you provide.

This is a good and valid point. We expanded this discussion, but we cannot pre-empt one of the results of our study that we use to revisit this aspect in the Discussion. Instead, as suggested in the next comment, we modified and expanded our statement in the subsequent paragraph.

Page 2, final paragraph: This paragraph begins with “thus, in order to optimize...” I would appreciate a bit more justification for this statement, as not all of your preceding examples lend themselves obviously to the conclusion that stabilizing selection on intermediate group sizes is to be expected (certainly the case in the yellow-bellied marmots and the anis, for instance, but it is less clear for the giraffe example). Perhaps this would simply require a bit of reordering between the preceding paragraph and this one to aid in the flow of logic from the discussion of fluctuating benefits across different environmental conditions to the expectation of an intermediate optimum.

See response to previous comment.

Page 3, paragraph 2: You mention including ASR and interannual variation in rainfall to put the present study in the context of other studies. This is understandable, but I would appreciate a little discussion here on why these variables might be considered biologically relevant to the present study species and whether the authors had any predictions as to how these factors might influence fitness outcomes in female sifakas.

We added a sentence for each of these variables to better explain why we included them in our analyses.

I also would recommend mentioning the species in question in the sentence “Furthermore, because a previous study suggested that female competition might impact reproductive rates in a sympatric species”. Further, are there any other similarities in social system between Verraeux’s sifakas and redfronted lemurs (i.e. female sympatry?) which would suggest a similar pressure acting in both species? If so, I would mention that, because at present it seems as though the only justification was that a sympatric species showed this pattern, which does not seem quite a sufficient justification on its own.

We made the suggested precision and added the suggested explanation.

Would the authors be able to clarify why the number of adult females vs. the ASR might yield different results, as was observed in your study?

We added a sentence at the end of the Introduction to explain why these two factors may have independent effects (ASR variation has stronger and more fundamental effects).

Discussion

Page 7: "While sifakas enjoyed higher survival in smaller groups and higher birth rates in groups with few other adult females, there are apparently no net fitness costs of living in small groups and no net fitness benefits of living in large groups, creating overall selective pressures favouring small groups" – this sentence makes it seem as though the second half is somehow in contradiction with the findings in the first clause, which do not appear to be the case. Both of the benefits presented in the first clause (higher survival in small groups and higher birth rates in groups with few other adult females, which also would be smaller groups by definition) suggest that all directionality of selection is going towards small groups, as you state. Therefore, a small rewarding might be beneficial.

We agree and deleted the perhaps confusing statement about potential costs of small group sizes.

Page nine, paragraph 2: "Group size does only play a subordinate role in predicting success in inter-group encounters 60, however, because proximity to their core area, rather than the numerical advantage of a group in a given encounter influenced the probability of winning" – I suggest re-wording this sentence for clarity.

Same paragraph: I think some of this justification for including intra-annual variation in rainfall and food availability could be relocated to the introduction.

We reworded this sentence for clarity and repeated the explanation for the significance of rainfall in this paragraph.

Page nine, final paragraph: missing a closing parentheses around the citation.

Added.

"Dominant males should therefore benefit reproductively from an increase in the number of adult females 67, revealing an apparent sexual conflict with females over group composition" – however, might there not be another side to this, in which males might be expected to benefit up to a certain point, after which they would not be able to

monopolize reproduction as effectively as they would like? In feral horses and zebras, which also have relatively small group sizes, males might be predicted to want larger harems only up until they are no longer able to defend paternity (e.g. Rubenstein & Nunez, 2009). I bring up equids because at least in terms of how much paternity a dominant male can monopolize and the proximate mechanisms that facilitate it (e.g. estrus asynchrony) appear to be similar, and group sizes also appear to be similar on average. The authors suggest that there would be an alignment between the dominant and subordinate male's interests, but I would argue that there might actually be three independent optima, in which females might prefer the smallest groups, subordinates might prefer the largest, and dominant males might prefer group sizes intermediate but perhaps closer to that of females than that of subordinate males. If this is the case, there could be a greater alignment between dominant males and females. I raise this as a potential alternative, and not necessarily mutually exclusive hypothesis for the author's consideration alongside the proposed mechanism.

This is an interesting thought, but we think it does not apply here because "the relatively small group sizes" might be absolutely too different. In primates, empirical studies revealed a critical number of adult females (i.e., 5-6) at which single males are no longer able to monopolize all females (Andelman SJ (1986) Ecological and social determinants of cercopithecine mating patterns. In: Rubenstein DI, Wrangham RW (eds) Ecological Aspects of Social Evolution: Birds and Mammals. Princeton University Press, Princeton, N.J., pp 201-216), but sifaka groups never even come close to that number. However, the reference to the equid study is relevant and interesting and we added a sentence to include it in our discussion here.

The authors suggests that there is a male and female conflict of interest and that female dominance wins in this intersexual conflict. I am curious if there is there any evidence to suggest that this conflict may be playing out in the form of males attempting to recruit or retain females and females attempting to prevent immigration or induce dispersal/eviction?

Good question. Unfortunately, we lack systematic data to answer it. However, we added a sentence to point out that female control over male immigration is limited because male infanticide has been observed repeatedly in our study population.

Page 10, paragraph 1: the authors mention more births and mothers were included in the present study compared with the previous one. How much larger was the sample size in the present study once these classification changes were implemented?

The present study included 8 more births by 3 more mothers.

Page 11, paragraph 1: "Also, survival and reproductive success seem to vary more often than not independently as a function of group size; i.e., different fitness components are evaluated independently." - I would suggest rewording for greater clarity.

Done.

Methods

Statistical and data collection methods appeared appropriate to the study aims.

Thanks!

Severine Hex

REVIEWERS' COMMENTS:

Reviewer #1 (Remarks to the Author):

I have now looked at the manuscript and believe that the authors have dealt with the comments

Reviewer #2 (Remarks to the Author):

I appreciate the work the authors put into integrating the reviewer's suggestions, as well as for the clarifications provided to justify decisions on analyses and structure. The additions made to the introduction and the discussion have added a great degree of context and clarity. I am satisfied with this version, and look forward to seeing it published.

Rebuttal letter

Thanks to both reviewers for their thumbs up. There were no additional request for changes from the editors. We implemented all required formal modifications indicted on the Editorial Requests Table.

REVIEWERS' COMMENTS:

Reviewer #1 (Remarks to the Author):

I have now looked at the manuscript and believe that the authors have dealt with the comments

Reviewer #2 (Remarks to the Author):

I appreciate the work the authors put into integrating the reviewer's suggestions, as well as for the clarifications provided to justify decisions on analyses and structure. The additions made to the introduction and the discussion have added a great degree of context and clarity. I am satisfied with this version, and look forward to seeing it published.